# FN-NOW: A Communication-Efficient Newton-Type Federated Learning via Low-Rank Hessian Approximation

## Abstract

Newton-type algorithms have become a promising direction for improving federated learning (FL). Their faster convergence offers new insights into enhancing communication efficiency in FL. However, these methods rely on the full Hessian, introducing significant computational, memory, and communication overhead. In this paper, we propose FN-NOW, a communication-efficient Newton-type federated optimization algorithm based on a low-rank approximation of the Hessian. Specifically, FN-NOW leverages Nyström method and the Woodbury identity to efficiently approximate the Hessian inverse, enabling communication-efficient training through fast convergence while maintaining memory overhead comparable to first-order methods. We provide a theoretical analysis showing that FN-NOW achieves a linear convergence rate under standard assumptions, outperforming typical first-order methods. Extensive experiments demonstrate that FN-NOW consistently outperforms existing methods in terms of both convergence speed and predictive performance, making it well suited for deployment in resource-constrained FL settings.

## 1 Introduction

Federated learning (FL) (McMahan et al., 2017) is a privacy-preserving distributed paradigm enabling collaborative model training across devices without sharing local data. However, frequent transmission of model parameters causes significant communication overhead (Martínez Beltrán et al., 2023; Liu et al., 2024a). Given bandwidth and network constraints, minimizing communication is essential for improving FL efficiency (Liu et al., 2024c). Since FedAvg (McMahan et al., 2017) introduced an SGD-based federated framework, numerous subsequent studies have aimed to further alleviate communication burdens (Zhao et al., 2023; Herzog et al., 2024). A common strategy in first-order methods is to reduce the size of local updates sent by clients (Konečnỳ, 2016; Chen et al., 2021; Di et al., 2024), which implicitly limits the amount of information aggregated.

Second-order optimization has gained attention for its fast convergence in centralized settings, with classical Newton's method (Cauchy, 1821; Fletcher & Powell, 1963) leveraging curvature information to accelerate training. This is appealing for FL, where fewer communication rounds are needed, potentially reducing communication costs. However, the vanilla Newton's method requires computing and transmitting a Hessian inverse with quadratic parameter complexity, imposing heavy resource demands and limiting scalability. Furthermore, this communication burden may offset the convergence gains. Therefore, making second-order methods practical for FL necessitates addressing the challenges of computational, memory, and per-round communication overhead.

Existing work has explored Newton-type optimizers in FL (Elgabli et al., 2022; Ma et al., 2022; Dinh et al., 2022), focusing on reducing communication and computation costs. Techniques include Hessian compression (Chaudhuri et al., 2022), Newton sketching (Li et al., 2024) and group alternating direction method of multipliers (ADMM) (Krouka et al., 2023). However, They still require storing full Hessian during local training, limiting applicability to shallow models with relatively few parameters. *This raises a natural question: can we retain the fast convergence of second-order methods in FL without their high overhead, achieving costs closer to first-order levels?*

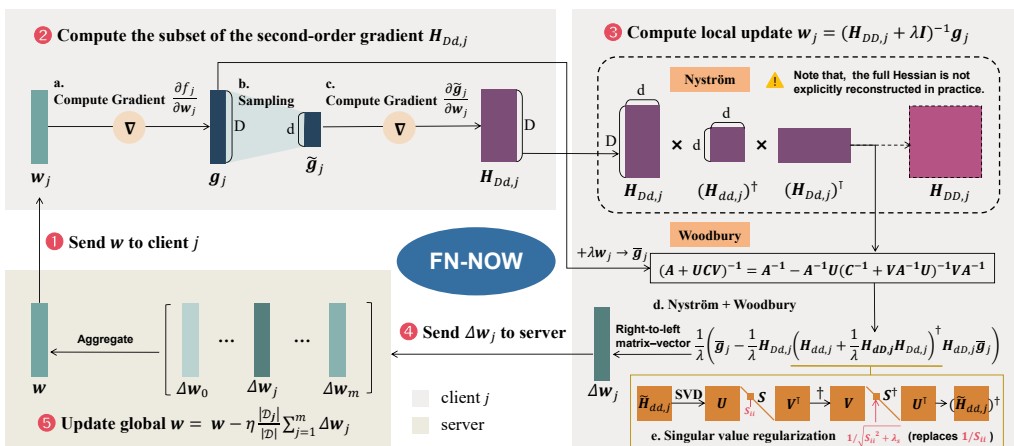

Figure 1: Illustration of FN-NOW.

Research on second-order optimizers for FL that are both resource-efficient and broadly applicable remains scarce. In this work, we aim to harness the advantages of second-order optimization while mitigating its costs in computation, memory, and per-round communication. Accordingly, we propose FN-NOW (Federated Newton's Method with Nyström and Woodbury), a novel federated Newton-type optimizer based on low-rank Hessian approximationas, illustrated in Figure 1. It replaces the full Hessian with Nyström approximation, and maintains low memory overhead with the Woodbury identity. By communicating parameter updates, it achieves per-round communication costs comparable to those of first-order methods. We provide theoretical and empirical evidence that FN-NOW converges faster, effectively reducing communication rounds. While primarily designed for communication efficiency, it also achieves strong accuracy and robustness to data heterogeneity. Notably, FN-NOW is not derived from existing centralized algorithms and is applicable to both federated and centralized settings. We summarize our main contributions as follows:

- We propose FN-NOW, a Newton-type FL method that reduces communication rounds via fast convergence and mitigates computation and memory overhead via Nyström approximation and the Woodbury identity, enabling scalability to diverse model architectures.

- We prove FN-NOW achieves a linear convergence rate under standard assumptions. This provides strong theoretical support for applying second-order methods in FL.

- Extensive experiments on benchmark datasets and commonly used models show that FN-NOW outperforms related methods in both convergence speed and accuracy, particularly in resource-constrained federated settings.

## 2 RELATED WORK

Our work focuses on applying second-order optimization to FL. We review existing federated optimizers and centralized second-order methods, which offer valuable insights for FL adaptation.

**Federated first-order optimizer.** McMahan et al. (2017) introduced FedAvg based on first-order optimizer SGD. Beyond SGD, Reddi et al. (2020) explored adaptive optimizers in FL like ADA-GRAD (Duchi et al., 2011), ADAM (Kingma, 2014), and YOGI (Zaheer et al., 2018), while Gong et al. (2022) proposed FedADMM using primal-dual optimization. Communication-efficient first-order FL methods typically either reduce per-device communication via compression (Chen et al., 2021) or filtering (Chu et al., 2025), or limit participating clients (Di et al., 2024; Ribero & Vikalo, 2024). Additionally, Herzog et al. (2024) reduces communication frequency by increasing local training. However, such strategies degrade aggregation quality and hinder overall training efficiency.

**Centralized second-order optimizer.** Classical techniques to accelerate computation include BFGS (Broyden, 1965), L-BFGS (Liu & Nocedal, 1989), Gauss-Newton (Schraudolph, 2002), and inexact Newton methods (Dembo et al., 1982). More recently, diagonal approximations have been

widely adopted. For example, ADAHESSIAN (Yao et al., 2021) and Sophia (Liu et al., 2024b) approximate the Hessian and Gauss-Newton (GN) diagonals, respectively. K-FAC (Martens & Grosse, 2015) represents the GN using a Kronecker product, and Botev et al. (2017) proposed recursive block-diagonal forms. HesScale (Elsayed et al., 2024) further enables scalable second-order updates with improved diagonal estimates. These methods offer key insights by approximating the Hessian with compact, informative substitutes.

**Federated second-order optimizer.** DANE (Shamir et al., 2014) and GIANT (Wang et al., 2018) use conjugate gradient to approximate Newton updates, while DONE (Dinh et al., 2022) applies Richardson iteration. These methods communicate twice per iteration. FedNL (Chaudhuri et al., 2022), SHED (Dal Fabbro et al., 2024) and FedNew (Elgabli et al., 2022) use compressed Hessians, eigenvector-eigenvalue pairs and ADMM mitigating communication, yet full Hessian is still required during training. FedNS (Li et al., 2024), FLeNS (Gupta et al., 2024) adopt sketched Hessians, but the limited compression ratio constrains their applicability across models. Recent works like Fed-Sophia (Elbakary et al., 2024), derived from centralized Sophia (Liu et al., 2024b), approximate the GN via diagonalization, while FAGH (Sen et al., 2024), samples the Hessian's first row. Their effectiveness hinges on accurately capturing the full Hessian. In comparison, our approach enables more adaptable approximation via flexible sampling. Moreover, we prove linear convergence, which is absent in existing methods relying on approximation.

## 3 PRELIMINARIES

**Federated learning.** We consider a standard FL setting with a global server and $m$ local devices. Each device $j \in [m]$ holds local data $\mathcal{D}_j$ drawn from a distribution $\rho_j$ over $\mathcal{X} \times \mathcal{Y}$ and full dataset is the disjoint union of local datasets $\mathcal{D} = \cup_{j=1}^m \mathcal{D}_j$, implicitly drawn from a global distribution $\rho$ over $\mathcal{X} \times \mathcal{Y}$, where $\mathcal{X}$ and $\mathcal{Y}$ denote the input and output spaces, respectively. The FL objective on $\mathcal{D}$:

$$\min_{\boldsymbol{w}} f(\boldsymbol{w}) = \sum_{j=1}^m \frac{|\mathcal{D}_j|}{|\mathcal{D}|} f_j(\boldsymbol{w}), \tag{1}$$

where $f_j(\boldsymbol{w})$ is the $j$-th device's loss function, $\boldsymbol{w} \in \mathbb{R}^D$ denotes the model parameters. A predefined learning rate $\eta$ is generally used. In centralized settings, gradient descent updates follow:

$$\boldsymbol{w}^t = \boldsymbol{w}^{t-1} - \eta \boldsymbol{g}^t, \tag{2}$$

where $\boldsymbol{g}^t := \nabla f(\boldsymbol{w}^t) \in \mathbb{R}^D$ is gradient. Let $\boldsymbol{w}^t$ denote the model at round $t$.

**Federated second-order optimizer.** Newton's method uses the inverse Hessian (second-order derivatives) to scale the gradient and determine the update direction:

$$\boldsymbol{w}^t = \boldsymbol{w}^{t-1} - \eta (\boldsymbol{H}^t)^{-1} \boldsymbol{g}^t, \tag{3}$$

where $\boldsymbol{H}^t := \nabla^2 f(\boldsymbol{w}^t) \in \mathbb{R}^{D \times D}$, and $\eta = 1$ recovers standard Newton's method.

In federated Newton's method, the global Hessian is obtained by aggregating local Hessian, analogous to gradient aggregation in first-order FL. For equation 3, they can be computed further as:

$$\boldsymbol{H}^t = \sum_{j=1}^m \frac{|\mathcal{D}_j|}{|\mathcal{D}|} \boldsymbol{H}_j^t, \quad \boldsymbol{g}^t = \sum_{j=1}^m \frac{|\mathcal{D}_j|}{|\mathcal{D}|} \boldsymbol{g}_j^t, \tag{4}$$

where $\boldsymbol{H}_j^t := \nabla^2 f_j(\boldsymbol{w}^t), \boldsymbol{g}_j^t := \nabla f_j(\boldsymbol{w}^t)$.

## 4 METHODOLOGY

As shown in Algorithm 1, we propose a Newton-type FL method via low-rank Hessian approximation to reduce computational and memory costs, while communicating updates at first-order level.

### 4.1 PROBLEM FORMULATION

Considering the addition of the $\ell_2$-norm of model parameters as a penalty term, the objective function of FL is given by:

$$\min_{\boldsymbol{w}} F(\boldsymbol{w}) = \sum_{j=1}^m \frac{|\mathcal{D}_j|}{|\mathcal{D}|} F_j(\boldsymbol{w}), \quad \text{where} \quad F_j(\boldsymbol{w}) = f_j(\boldsymbol{w}) + \frac{\lambda}{2} \|\boldsymbol{w}\|^2, \tag{5}$$

---

**Algorithm 1** FN-NOW

---

**Input**: Local training data subset $\mathcal{D}_j$, $\forall j \in [m]$, local loss function $F_j(\boldsymbol{w})$, number of communication rounds $T$, the number of local examples $|\mathcal{D}_j|$ and total examples $|\mathcal{D}|$.
**Parameter**: Learning rate $\eta$, regularization parameters $\lambda$, stabilization parameter $\lambda_s$.
**Output**: The global weight $\boldsymbol{w}$

1: **for** each round $t = 1, \ldots, T$ **do**
2:    Communicate $\boldsymbol{w} = (x_1, \ldots, x_D)^\top$ to all clients.
3:    **for** each client $j \in [m]$ **do**
4:       $\boldsymbol{w}_j \leftarrow \boldsymbol{w}$.
5:       **for** each step (batches in each epoch) **do**
6:          Compute $\boldsymbol{g}_j \leftarrow \frac{\partial f_j}{\partial \boldsymbol{w}_j}$, $\bar{\boldsymbol{g}}_j \leftarrow \boldsymbol{g}_j + \lambda \boldsymbol{w}_j$.
7:          Sample gradient subset $\widetilde{\boldsymbol{g}}_j \subseteq \boldsymbol{g}_j$ via leverage score sampling in equation 8
8:          Compute $\boldsymbol{H}_{Dd,j} \leftarrow \frac{\partial \widetilde{\boldsymbol{g}}_j}{\partial \boldsymbol{w}_j}$, $\boldsymbol{H}_{dd,j} \subseteq \boldsymbol{H}_{Dd,j}$ and $\widetilde{\boldsymbol{H}}_{dd,j} = \boldsymbol{H}_{dd,j} + \frac{1}{\lambda} \boldsymbol{H}_{dD,j} \boldsymbol{H}_{Dd,j}$
9:          Perform SVD for the matrix $\widetilde{\boldsymbol{H}}_{dd,j} = \boldsymbol{U}\boldsymbol{S}\boldsymbol{V}^\top$.
10:        Apply singular value regularization to compute the inverse of $(\widetilde{\boldsymbol{H}}_{dd,j})^\dagger = \boldsymbol{V}\widetilde{\boldsymbol{S}}^+\boldsymbol{U}^\top$ according to equation 11, where $\widetilde{\boldsymbol{S}}^+$ is a diagonal matrix with $\widetilde{S}_{ii}^+ = 1/\sqrt{S_{ii}^2 + \lambda_s}$ and $\lambda_s > 0$.
11:        Compute local update $\Delta \boldsymbol{w}_j = \frac{1}{\lambda}\left(\bar{\boldsymbol{g}} - \frac{1}{\lambda}\left(\boldsymbol{H}_{Dd,j}\left(\left(\widetilde{\boldsymbol{H}}_{dd,j}\right)^\dagger(\boldsymbol{H}_{dD,j}\bar{\boldsymbol{g}}_j)\right)\right)\right)$ according to equation 10 with matrix-vector multiplications.
12:       **end for**
13:       Communicate the local update $\Delta \boldsymbol{w}_j$ to server.
14:    **end for**
15:    **On the global server:**
16:    Update the global model $\boldsymbol{w} \leftarrow \boldsymbol{w} - \eta \sum_{j=1}^m \frac{|\mathcal{D}_j|}{|\mathcal{D}|} \Delta \boldsymbol{w}_j$.
17: **end for**

---

where $\lambda$ is a regularization parameter. The solution using Newton's method can be expressed as:

$$\boldsymbol{w}^t = \boldsymbol{w}^{t-1} - \eta \frac{|\mathcal{D}_j|}{|\mathcal{D}|} \sum_{j=1}^m \Delta \boldsymbol{w}_j^{t-1}, \quad \Delta \boldsymbol{w}_j^{t-1} = (\boldsymbol{H}_j^{t-1} + \lambda \boldsymbol{I})^{-1} \bar{\boldsymbol{g}}_j^{t-1}, \tag{6}$$

where $\bar{\boldsymbol{g}}_j^t := \boldsymbol{g}_j^t + \lambda \boldsymbol{w}^t$, and the remaining notation is consistent with equation 3.

## 4.2 Hessian Matrix with Nyström Aprroximation

The Nyström method (Williams & Seeger, 2000) is an approximation technique used to efficiently handle large-scale matrices by utilizing a subset of columns. This means that in second-order methods, we can approximate the entire Hessian matrix by computing only a little part of it. We consider training on a certain device where $\boldsymbol{w} = (x_1, \ldots x_D)^\top \in \mathbb{R}^D$ represents the model parameters to be optimized, and the first-order gradient $\boldsymbol{g} = (g_1, \ldots, g_D)^\top \in \mathbb{R}^D$, where $g_i = \frac{\partial f}{\partial x_i}$, can be easily computed. Then we sample the subset $\widetilde{\boldsymbol{g}} = (\widetilde{g}_1, \ldots, \widetilde{g}_d)^\top \subseteq \boldsymbol{g}$, $d \ll D$ to calculate the subset of the second-order gradient $\boldsymbol{H}_{Dd} := (\frac{\partial \widetilde{\boldsymbol{g}}}{\partial x_1}, \ldots, \frac{\partial \widetilde{\boldsymbol{g}}}{\partial x_D})^\top \in \mathbb{R}^{D \times d}$. Using Nyström technique, we approximate the Hessian matrix and the local update in Newton's method according to equation 6:

$$\boldsymbol{H}_{DD} \approx \boldsymbol{H}_{Dd}(\boldsymbol{H}_{dd})^\dagger \boldsymbol{H}_{Dd}^\top, \quad (\boldsymbol{H}_{DD} + \lambda \boldsymbol{I})^{-1} \bar{\boldsymbol{g}} \approx (\boldsymbol{H}_{Dd}(\boldsymbol{H}_{dd})^\dagger \boldsymbol{H}_{Dd}^\top + \lambda \boldsymbol{I})^{-1} \bar{\boldsymbol{g}}, \tag{7}$$

where $\boldsymbol{H}_{DD} := (\frac{\partial \boldsymbol{g}}{\partial x_1}, \ldots, \frac{\partial \boldsymbol{g}}{\partial x_D})^\top \in \mathbb{R}^{D \times D}$, $\boldsymbol{H}_{dd} \in \mathbb{R}^{d \times d} \subseteq \boldsymbol{H}_{Dd}$, $\boldsymbol{H}^\dagger$ is the Moore-Penrose inverse of the martrix $\boldsymbol{H}$.

Since the Hessian matrix may be sparse, uniform sampling can yield poorly informative subsets, leading to unstable or inaccurate inversions. Sampling more informative components improves the approximation quality. We use leverage score sampling method to quantify the importance of data points in the regularized kernel matrix. In our method, the subset $\widetilde{\boldsymbol{g}}$ is selected with probability:

$$p_i = \widehat{l}(i) / \sum_{i=1}^D \widehat{l}(i), \quad \widehat{l}(i) = g_i^2 / \sum_{i=1}^D g_i^2, \tag{8}$$

where $\widehat{l}(i)$ is the leverage score of $g_i \in \boldsymbol{g}$ and $i \in [D]$.

### 4.3 Inverse Hessian with Woodbury Identity

Although we have introduced low-rank approximation techniques for Hessian, in Newton's method, the gradient update is actually performed using the inverse of the Hessian matrix. It is necessary to avoid the constructing of full-sized Hessian matrices throughout the entire computation process. The Woodbury identity (Sherman & Morrison, 1950) simplifies the computation of matrix inverses through matrix decomposition and is applicable when certain parts of the matrix have a special structure. The identity is as follows:

$$(\boldsymbol{A} + \boldsymbol{U}\boldsymbol{C}\boldsymbol{V})^{-1} = \boldsymbol{A}^{-1} - \boldsymbol{A}^{-1}\boldsymbol{U}(\boldsymbol{C}^{-1} + \boldsymbol{V}\boldsymbol{A}^{-1}\boldsymbol{U})^{-1}\boldsymbol{V}\boldsymbol{A}^{-1}, \tag{9}$$

where $\boldsymbol{A} \in \mathbb{R}^{N \times N}$ is typically a large and sparse matrix, $\boldsymbol{U} \in \mathbb{R}^{N \times n}, \boldsymbol{C} \in \mathbb{R}^{n \times n}, \boldsymbol{V} \in \mathbb{R}^{n \times N}$, and $n \ll N$.

Using the Woodbury identity equation 9 to decompose the inverse of the Hessain matrix equation 7, the second-order update computed on local devices can be decomposed as:

$$(\boldsymbol{H}_{DD} + \lambda \boldsymbol{I})^{-1}\bar{\boldsymbol{g}} \approx \frac{1}{\lambda}\left(\bar{\boldsymbol{g}} - \frac{1}{\lambda}\boldsymbol{H}_{Dd}\left((\widetilde{\boldsymbol{H}}_{dd})^{\dagger}(\boldsymbol{H}_{dD}\bar{\boldsymbol{g}})\right)\right). \tag{10}$$

We denote $\boldsymbol{H}_{dD} := \boldsymbol{H}_{Dd}^{\top}, \widetilde{\boldsymbol{H}}_{dd} := \boldsymbol{H}_{dd} + \frac{1}{\lambda}\boldsymbol{H}_{dD}\boldsymbol{H}_{Dd}$ and since the matrix may be singular, we use the Moore-Penrose pseudoinverse to replace the matrix inverse.

This differs from previous federated second-order methods, which typically communicate both the Hessian matrix and gradients. In contrast, our approach, which only communicates the local parameter updates, reduces the communication overhead to the level of first-order methods. We demonstrate the effectiveness of directly aggregating parameter updates through both experimental results and theoretical analysis.

**Remark 1** (Efficient Matrix Inversion and Matrix-Vector Multiplication). *The computation in equation 7 involves the inverse of a $D \times D$ matrix, resulting in a time complexity of $\mathcal{O}(D^3)$, which is prohibitive for high-dimensional models. By applying the Woodbury matrix identity, this inversion can be reformulated in terms of a much smaller $d \times d$ matrix in equation 10, where $d \ll D$, significantly reducing the computational burden. Furthermore, to avoid expensive matrix-matrix multiplications, we compute the local update through a sequence of matrix-vector multiplications. After the use of the Woodbury identity and matrix-vector multiplications, the computational complexity is reduced from $\mathcal{O}(D^3)$ in equation 7 to $\mathcal{O}(Dd + d^3)$ in equation 10, where $d \ll D$.*

### 4.4 Singular Value Regularization

We compute the pseudo-inverse of $\boldsymbol{H}_{dd}$ using SVD, a standard and stable approach, and conveniently leverage its structure to perform singular value regularization. The Moore-Penrose pseudoinverse of $\boldsymbol{A}$ is obtained by decomposing it as $\boldsymbol{U}\boldsymbol{S}\boldsymbol{V}^{\top}$, inverting the nonzero singular values in $\boldsymbol{S}$ to form $\boldsymbol{S}^{+}$, and reconstructing $\boldsymbol{V}\boldsymbol{S}^{+}\boldsymbol{U}^{\top}$. The pseudoinverse in equation 10, incorporating regularization, is computed as:

$$(\widetilde{\boldsymbol{H}}_{dd})^{\dagger} = \boldsymbol{V}\widetilde{\boldsymbol{S}}^{+}\boldsymbol{U}^{\top}, \tag{11}$$

where $\widetilde{\boldsymbol{S}}^{+}$ is the diagonal matrix with $\widetilde{S}_{ii}^{+} = 1/\sqrt{S_{ii}^2 + \lambda_s}$ and $\lambda_s > 0$ is stabilization parameter.

**Remark 2** (Stability of Hessian Inversion). *Hessian inversion can be unstable due to near-zero eigenvalues, and Liu et al. (2024b) further pointed out issues from rapidly changing or negative curvature. Therefore, Hessian conditioning is essential in second-order methods. For example, Liu et al. (2024b) and Elbakary et al. (2024) impose a lower bound on second-order information and clips update to ensure stability. We regularize the singular values in the SVD step of the Moore–Penrose pseudoinverse computation to directly address the core instability without altering curvature directions while making explicit use of the existing decomposition.*

## 5 Convergence Analysis

In this section, we provide a convergence analysis of FN-NOW and a theoretical comparison with several methods. Before the analysis, we first introduce some notations and standard assumptions.

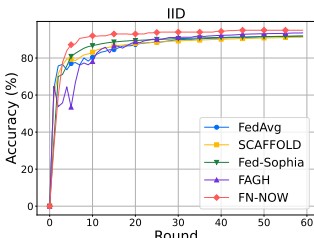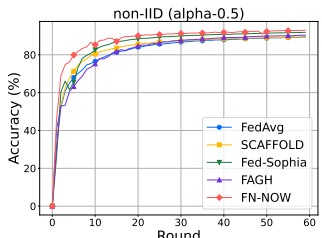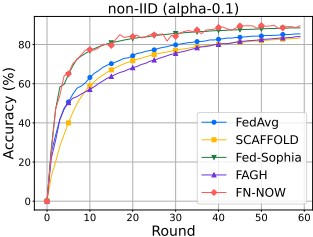

Figure 2: The test accuracy of the compared methods on MNIST using MLP under different levels of data heterogeneity.

**Assumption 1** (Twice differentiable and convex). *The objective function are closed and twice differentiable convex function and $\nabla^2 F(\boldsymbol{w}) \succeq \upsilon \boldsymbol{I}$.*

**Assumption 2** (Lipschitz smoothness). *The second-order derivatives of $F$ are $M$-Lipschitz smooth, i.e., $\|\nabla^2 F(\boldsymbol{w}) - \nabla^2 F(\boldsymbol{w}')\| \leq M\|\boldsymbol{w} - \boldsymbol{w}'\|$. The gradient satisfies the $L$-lipschitz condition, i.e., $\|\nabla F(\boldsymbol{w}) - \nabla F(\boldsymbol{w}')\| \leq L\|\boldsymbol{w} - \boldsymbol{w}'\|$.*

**Lemma 1** (Nyström approximation error). *With Assumptions 1, 2, let $c > 0$ and $\delta \in (0,1)$ be fixed constants. For a target accuracy $\epsilon_l \in (0,1)$ and target rank $k$ of the low-rank approximation, the sampling dimension $d \geq \frac{c}{\epsilon_l^2} k \ln \frac{k}{\delta}$, the exact Hessian $\boldsymbol{H}$ and its Nyström approximation $\hat{\boldsymbol{H}}$ satisfy*

$$\|\boldsymbol{H} - \hat{\boldsymbol{H}}\| \leq \rho_{Ny}, \tag{12}$$

*where $\rho_{Ny} = (1 + \epsilon_l)\lambda_{k+1}(\boldsymbol{H})$, $\lambda_{k+1}(\boldsymbol{H})$ denotes the $(k+1)$-th eigenvalue of $\boldsymbol{H}$, with probability at least $1 - \delta$. A detailed proof is given in A.2.*

### 5.1 CONVERGENCE ANALYSIS FOR FN-NOW

**Theorem 1.** *(Convergence of FN-NOW). Under Assumptions 1, 2, let $\delta \in (0,1)$ and $\epsilon \in (0, \frac{1}{2})$. Suppose that each $\nabla^2 F_j(\cdot)$ is uniformly upper bounded, i.e., $\nabla^2 F_j(\boldsymbol{w}) \preceq C\boldsymbol{I}$ for all $j \in [m]$. When $|\mathcal{D}_j| \geq \frac{4C}{\upsilon \epsilon^2} \log \frac{2DK}{\delta}$ and $d \geq \frac{c}{\epsilon_l^2} k \ln \frac{kK}{\delta}$, we obtain*

$$\|\boldsymbol{w}^{t+1} - \boldsymbol{w}^*\| \leq P\|\boldsymbol{w}^t - \boldsymbol{w}^*\| + \frac{3M}{2\upsilon}\|\boldsymbol{w}^t - \boldsymbol{w}^*\|^2, \tag{13}$$

*with probability at least $1 - 3K\delta$. Here, $P$ is a constant, defined as $P = \frac{(1-\eta)L}{\upsilon} + \frac{\eta\Gamma L}{(1-\epsilon)\upsilon^2} + \frac{\eta\rho_{Ny}L}{(1-\epsilon)\upsilon\lambda} + \frac{\eta\epsilon_B L}{\lambda}$, where $\Gamma$ and $\epsilon_B$ are related to the local similarity of the first and second order gradients, respectively.*

In equation 13, the algorithm converges when $P < 1$, and the accompanying discussion and detailed proof are provided in B. We now analyze the components of $P$. The first term captures the deviation from the standard Newton method. This term vanishes when $\eta = 1$, under ideal conditions. The second and fourth terms depend on distributional differences across local clients and increase with the degree of heterogeneity. To quantify distributional dissimilarity, we adopt a commonly used similarity bound defined in (Li et al., 2020; Karimireddy et al., 2020). The third term reflects the error introduced by the Nyström approximation, which depends on sampling quality and the number of sampled columns.

**Theorem 2.** *(Convergence rate of FN-NOW). Under Assumptions 1, 2 and iterative process in Theorem 1, if initial point satisfies $\|\boldsymbol{w}^0 - \boldsymbol{w}^*\| \leq \frac{(1-P)\upsilon}{M}$, then achieving $\|\boldsymbol{w}^t - \boldsymbol{w}^*\| \leq \varepsilon$ requires $T = \mathcal{O}(\log \frac{1}{\varepsilon})$ iterations.*

The proof is in C. This result establishes linear convergence, as typical for Newton-type methods. Although the second-order term is not globally dominant, it improves convergence near the optimum, often after a few warm-up steps with a first-order method.

### 5.2 THEORETICAL COMPARISONS

We theoretically compare our method with the ideal Newton's method and other related methods in Table 1. Compared to first-order methods FedAvg (McMahan et al., 2017) and FedProx (Dinh et al.,

Table 1: Summary of Communication Complexity (Comm.) and Memory Overhead Comparison for Related Methods.

| METHOD | MEMORY COST | ITERATIONS $T$ | COMM. ONCE | COMM. COMPLEXITY |
|---|---|---|---|---|
| FEDAVG | $\mathcal{O}(D)$ | $\mathcal{O}(\frac{1}{\varepsilon})$ | $\mathcal{O}(D)$ | $\mathcal{O}(\frac{D}{\varepsilon})$ |
| FEDPROX | $\mathcal{O}(D)$ | $\mathcal{O}(\frac{1}{\varepsilon})$ | $\mathcal{O}(D)$ | $\mathcal{O}(\frac{D}{\varepsilon})$ |
| DANE | $\mathcal{O}(D)$ | $\mathcal{O}(\frac{\kappa^2}{|\mathcal{D}_j|}\log(Dm)\log\frac{1}{\varepsilon})$ | $\mathcal{O}(D)$ | $\mathcal{O}(D\frac{k^2}{|\mathcal{D}_j|}\log(Dm)\log\frac{1}{\varepsilon}))$ |
| DONE | $\mathcal{O}(D^2)$ | $\mathcal{O}(\delta_\kappa\log\frac{1}{\varepsilon})$ | $\mathcal{O}(D)$ | $\mathcal{O}(D\delta_k\log\frac{1}{\varepsilon})$ |
| | | $\mathcal{O}(\log\log\frac{1}{\varepsilon})$ | $\mathcal{O}(D)$ | $\mathcal{O}(D\log\log\frac{1}{\varepsilon})$ |
| FEDNL | $\mathcal{O}(D^2)$ | $\mathcal{O}(\log\frac{1}{\varepsilon})$ | $\mathcal{O}(D)$ | $\mathcal{O}(D\log\frac{1}{\varepsilon})$ |
| SHED | $\mathcal{O}(D^2)$ | $\mathcal{O}(\log\frac{1}{\varepsilon})$ | $-$ | $\mathcal{O}(D^2)$ |
| FEDNEWTON | $\mathcal{O}(D^2)$ | $\mathcal{O}(\log\log\frac{1}{\varepsilon})$ | $\mathcal{O}(D^2)$ | $\mathcal{O}(D^2\log\log\frac{1}{\varepsilon})$ |
| **FN-NOW** | $\mathcal{O}(dD)$ | $\mathcal{O}(\log\frac{1}{\varepsilon})$ | $\mathcal{O}(D)$ | $\mathcal{O}(D\log\frac{1}{\varepsilon})$ |

NOTE: ALL ANALYSES ARE CARRIED OUT UNDER THE ASSUMPTION OF A $\varepsilon$-ACCURATE SOLUTION. DONE AND DANE ACTUALLY COMMUNICATE TWICE PER ROUND. $\kappa$ IN ITERATIONS OF DANE REPRESENTS THE CONDITION NUMBER. $\delta_k < 1$ IS ASSUMED IN DONE

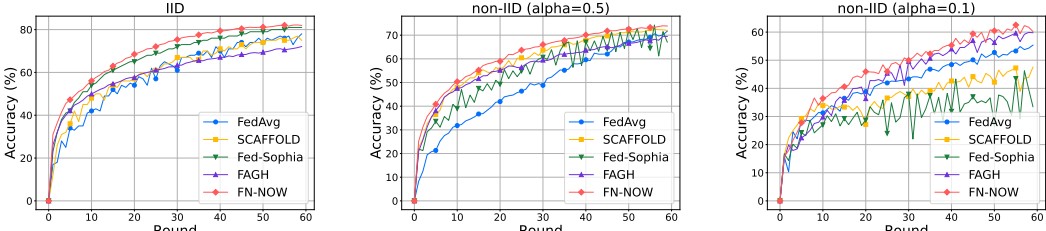

Figure 3: The test accuracy of the compared methods on CIFAR10 using ResNet-18 under different levels of data heterogeneity.

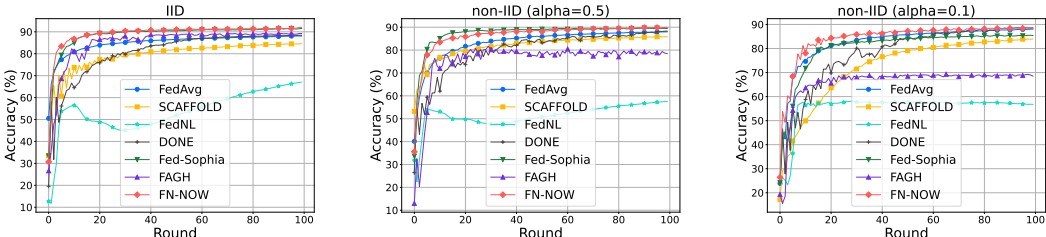

Figure 4: The test accuracy of the compared methods on MNIST using MLR under different levels of data heterogeneity.

2022), second-order methods require fewer iterations $T$, and our method retains this advantage without additional communication overhead. Other second-order methods, including Done (Dinh et al., 2022), FedNL (Chaudhuri et al., 2022), and SHED (Dal Fabbro et al., 2024), have low communication complexity but incur high memory overhead. DANE (Shamir et al., 2014) is memory-efficient but communication-inefficient.

Recent methods such as FAGH (Sen et al., 2024) and Fed-Sophia (Elbakary et al., 2024) retain the Newton update form and have low memory overhead, but lack theoretical guarantees, and are thus analyzed algorithmically. FAGH, approximating the Hessian by sampling its first row, is a special case of the Nyström method. It corresponds to our method with sampling size 1, while ours allows greater flexibility. Fed-Sophia uses a diagonal approximation with per-iteration complexity $\mathcal{O}(D)$, matching ours. Due to algorithmic differences, we compare convergence empirically.

Table 2: The comparison of communication cost (Comm.) and the number of rounds required by the compared methods to reach a target accuracy (as shown in the third row of the table), with experimental settings consistent with those described earlier.

| METHOD | TARGET ACCURACY - MLP | | | | | | TARGET ACCURACY - RESNET | | | | | |
|---|---|---|---|---|---|---|---|---|---|---|---|---|
| | COMM. (MB) | IID | | $\alpha = 0.5$ | | $\alpha = 0.1$ | COMM. (MB) | IID | | $\alpha = 0.5$ | | $\alpha = 0.1$ |
| | | 80 | 90 | 80 | 90 | 80 | 85 | 60 | 70 | 60 | 70 | 40 | 50 |
| FEDAVG | 45.43 | 7 | 31 | 13 | - | 30 | 54 | 1279.64 | 23 | 39 | 21 | 55 | 21 | 47 |
| SCAFFOLD | 90.87 | 6 | 37 | 9 | - | 39 | - | 2559.27 | 21 | 35 | 23 | 45 | 36 | - |
| FED-SOPHIA | 45.43 | 4 | 24 | 8 | 30 | 13 | 26 | 1279.64 | 13 | 26 | 28 | 47 | 40 | - |
| FAGH | 90.87 | 10 | 23 | 13 | 52 | 39 | - | 2559.27 | 22 | 51 | 32 | - | 17 | 32 |
| **FN-NOW** | 45.43 | **3** | **6** | **5** | **19** | **12** | **23** | 1279.64 | **12** | **21** | **21** | **39** | **13** | **28** |

# 6 EXPERIMENTS

To evaluate FN-NOW, we conducted experiments on MNIST, Fashion MNIST, and CIFAR-10, using three models of increasing complexity: a single-layer MLP, a five-layer CNN (1.5M parameters), and ResNet-18 (He et al., 2015). We compared against first-order methods FedAvg (McMahan et al., 2017) and SCAFFOLD (Karimireddy et al., 2020), and second-order methods Fed-Sophia (Elbakary et al., 2024), FAGH (Sen et al., 2024), DONE (Dinh et al., 2022), and FedNL (Chaudhuri et al., 2022), with the latter two evaluated on multinomial logistic regression (MLR). Data heterogeneity was simulated via a Dirichlet distribution $\pi \sim \mathrm{Dir}_r(\alpha)$, where larger $\alpha$ implies more IID. Hyperparameters were selected based on convergence speed, final accuracy and the recommended settings in original paper. All results are averaged over five runs. Full experimental details are provided in D.

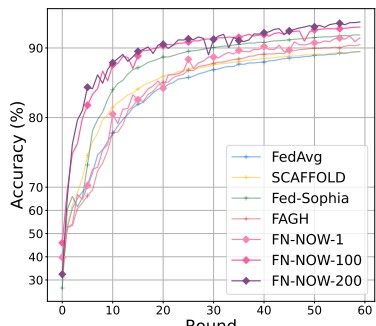

Figure 5: The test accuracy of FN-NOW with different sample scale and other compared methods. The number following the method name represents the size of samples, and the total number of parameters in the model is 392,000.

**Overall training performance.** We evaluate training performance across methods with 30 clients by comparing test accuracy over communication rounds under three heterogeneity levels: IID, moderate ($\alpha = 0.5$), and extreme ($\alpha = 0.1$). As shown in Figures 2–4, our method generally converges faster, occasionally matching Fed-Sophia under extreme heterogeneity but still outperforming other baselines. FedAvg performs best on MLR, likely due to the simplicity of the model where SGD suffices and complex methods may hinder performance. FedNL performs poorly due to large errors from its crude compression. These results highlight the advantage of efficient optimizers in complex settings, consistent with our goal of scalable second-order methods. Among Hessian approximation methods, FAGH suffers from slow convergence and sharp degradation as heterogeneity increases, likely due to loss of Hessian information. In contrast, Fed-Sophia and our method benefit from richer sampling. On ResNet, the most complex model, our method outperforms Fed-Sophia with similar computational cost, likely due to better preservation of feature interactions. CNN results are in the E.1; partial client participation settings are in E.2.

**Impact of sample scale.** We evaluated the impact of Nyström sampling size on non-IID ($\alpha = 0.5$) MNIST using an MLP. As shown in Figure 5, performance significantly improves from size 1 to 100, while further increasing from 100 to 200 offers marginal gains. This suggests limited additional information from larger samples, indicating FN-NOW achieves strong performance without incurring unnecessary memory or computational overhead.

**communication efficiency.** Table 2 reports the per-round communication cost and the rounds required to first reach target accuracy. MLR and CNN results appear in E.3. The table shows our method lowers both rounds and communication once. Figure 6 further presents training wall clock time and communication time for MLP with non-IID data ($\alpha = 0.5$), computing communication time from the measured round count under a fixed 10 Mbps link rate following (Chen et al., 2023). The figure underscores the importance of communication efficiency. Although larger sampling

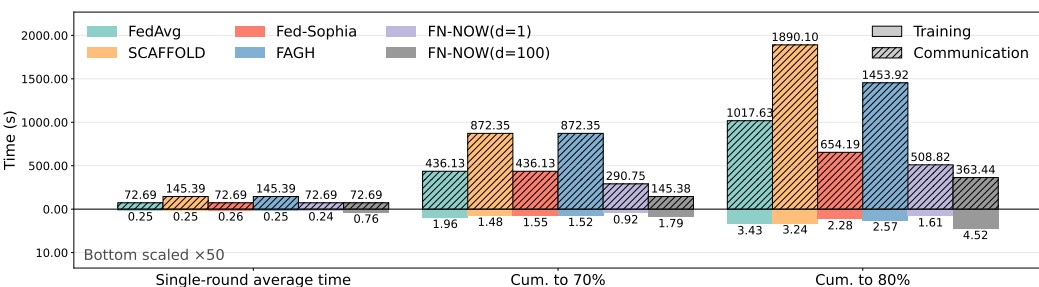

Figure 6: Average per-round training/communication time, and cumulative training/communication time until accuracy first reaches 70% (Cum. to 70%) and 80% (Cum. to 80%).

Table 3: The summary of final-round accuracy (%) for the compared methods, with experimental settings consistent with those described earlier.

| MODEL | | FEDAVG | SCAFFOLD | FED-SOPHIA | FAGH | **FN-NOW** |
|---|---|---|---|---|---|---|
| MLP | IID | 91.86±0.02 | 91.31±0.04 | 91.87±0.04 | 91.86±0.02 | **95.00±0.04** |
| | $\alpha = 0.5$ | 89.48±0.13 | 89.47±0.04 | 89.48±0.13 | 90.47±0.11 | **93.00±0.05** |
| | $\alpha = 0.1$ | 85.52±1.49 | 83.38±0.24 | 89.31±0.11 | 84.17±0.41 | **89.72±0.53** |
| CNN | IID | 84.05±0.91 | 79.60±1.96 | 87.31±0.24 | 80.40±0.42 | **87.89±0.22** |
| | $\alpha = 0.5$ | 82.86±0.28 | 77.32±0.33 | 85.01±0.19 | 83.01±0.03 | **86.11±0.18** |
| | $\alpha = 0.1$ | 74.49±1.50 | 75.51±0.59 | 82.45±1.03 | 75.08±0.11 | **83.36±0.32** |
| RESNET | IID | 78.05±0.05 | 76.35±0.06 | 81.87±1.54 | 72.19±0.13 | **82.12±0.79** |
| | $\alpha = 0.5$ | 71.95±0.32 | 72.87±0.78 | 71.46±3.53 | 69.07±0.19 | **73.96±0.43** |
| | $\alpha = 0.1$ | 54.40±1.76 | 46.99±1.75 | 39.83±4.49 | 60.84±1.58 | **61.03±0.87** |

slightly lengthens training time for our method, the reduction in communication time dominates. Overall, by reducing both the number of rounds and the per-round payload, FN-NOW improves communication efficiency and thus overall efficiency.

**Model Accuracy.** We compare the final accuracy at the last round 60 in Table 3. Despite being primarily designed to improve communication efficiency, our method also delivers strong accuracy and robustness to data heterogeneity. These results validate second-order methods as an effective approach to improving communication efficiency in FL without sacrificing training quality. Additional results and detailed analysis on MLR are provided in E.4.

**The impact of regularization parameter $\lambda$.** In Figure 7, we assess $\lambda$ on a CNN trained on Fashion MNIST with non-IID data ($\alpha = 0.5$). Larger $\lambda$ stabilizes opti-

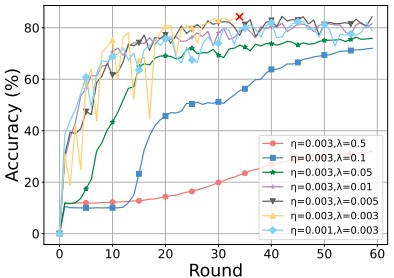

Figure 7: Training comparison across $\lambda$.

mization but degrades performance; smaller $\lambda$ induces instability, and below a threshold training aborts due to ill-conditioned inversions. This aligns with the role of $\lambda$ as a regularizer that enforces Hessian positive definiteness—larger values improve conditioning but attenuate useful second-order information. We also observe coupling with the learning rate $\eta$; when training fails at very small $\lambda$, reducing $\eta$ restores stability.

## 7 CONCLUSION

Second-order optimization in FL faces high computational, memory, and per-round communication costs. We present FN-NOW, a Newton-type algorithm that retains second-order benefits while significantly reducing overhead through Nyström approximation and the Woodbury identity, and provide theoretical guarantees of linear convergence. FN-NOW approaches first-order methods in memory and communication, and achieves substantial computational savings over standard second-order methods, though further improvements remain possible. Future work may explore trade-offs between performance and cost under flexible sampling.

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

## A  AUXILIARY LEMMAS

### A.1  LOCAL SECOND-ORDER GRADIENT BOUND (FROM LEMMA A.1. IN GUPTA ET AL. (2021))

We directly adopt the result from Lemma A.1. in Gupta et al. (2021). Under Assumptions 1, 2, and $\nabla F_j^2(\boldsymbol{w}) \preceq C\boldsymbol{I}$, let $\epsilon \in (0, \frac{1}{2})$ and $\delta \in (0,1)$. Then, for a given $|\mathcal{D}_j| \geq \frac{4C}{v\epsilon^2} \log \frac{2D}{\delta}$, we obtain

$$(1 - \epsilon)v \preceq \nabla^2 F_j(\boldsymbol{w}) \preceq (1 + \epsilon)L, \tag{14}$$

for all $\boldsymbol{w} \in \mathbb{R}^D$ and $j \in [m]$ with probability at least $1 - \delta$.

### A.2  PROOF OF LEMMA 1

*Proof.* The approximation error of the Nyström has been extensively studied in previous work. Here, we introduce a result from Gittens & Mahoney (2013) concerning uniform sampling. Let $\boldsymbol{H} \in \mathbb{R}^{D \times D}$ be a symmetric positive definite matrix with eigendecomposition $\boldsymbol{H} = \sum_{i=1}^{D} \lambda_i \boldsymbol{u}_i \boldsymbol{u}_i^\top$, where $\{\lambda_i\}_{i=1}^{D}$ are the eigenvalues and $\{\boldsymbol{u}_i\}_{i=1}^{D}$ are the corresponding orthonormal eigenvectors of $\boldsymbol{H}$. Let $\boldsymbol{H}_k = \sum_{i=1}^{k} \lambda_i \boldsymbol{u}_i \boldsymbol{u}_i^\top$ denote the best rank-$k$ approximation of $\boldsymbol{H}$. Then, there exists a constant $c > 0$ and a parameter $\beta = \frac{D}{k} \max_i \hat{l}(i)$ that measures the distribution of the column space, such that when $d \geq c\beta k \ln \frac{k}{\delta}$, the approximation error between $\boldsymbol{H}$ and its low-rank approximation $\hat{\boldsymbol{H}}$ is bounded by

$$\|\boldsymbol{H} - \hat{\boldsymbol{H}}\| \leq (1 + \frac{2D}{d})\|\boldsymbol{H} - \boldsymbol{H_k}\|, \tag{15}$$

with probability at least $1 - \delta$ and where $\|\boldsymbol{H} - \boldsymbol{H_k}\| = \|\sum_{i=k+1}^{D} \lambda_i \boldsymbol{u_i} \boldsymbol{u_i}^\top\| = \lambda_{k+1}(\boldsymbol{H})$.

Building on this, we consider the case of leverage score sampling. Unlike uniform sampling, which requires a larger number of samples to ensure quality, leverage score sampling focuses solely on approximation accuracy and does not depend on data distribution. Setting $\epsilon_l \in (0,1)$ and applying the Matrix Bernstein inequality, we obtain the following result when $d \geq \frac{c}{\epsilon_l^2} k \ln \frac{k}{\delta}$:

$$\|\boldsymbol{H} - \hat{\boldsymbol{H}}\| \leq (1 + \epsilon_l)\lambda_{k+1}(\boldsymbol{H}) = \rho_{Ny}, \tag{16}$$

with probability at least $1 - \delta$. Using a larger $d$ allows the approximation error $\epsilon_l$ to be reduced, leading to improved approximation quality, which is consistent with both our empirical observations and experimental results. Moreover, leverage score sampling reduces the amplification factor $1 + \frac{D}{d}$ to a controllable level determined by the target precision.

$\square$

## B  PROOF OF THEOREM 1

*Proof.* We analyze the full-batch case with one training epoch. Intuitively, the entire update can be decomposed into the ideal global exact update and an error term, which can then be analyzed separately. Before proceeding with the proof, we define $\bar{\boldsymbol{g}}^t := \nabla F(\boldsymbol{w}^t)$, $\bar{\boldsymbol{g}}_j^t := \nabla F(\boldsymbol{w}_j^t)$, $\boldsymbol{H}_F^t := \nabla^2 F(\boldsymbol{w}^t)$, $\boldsymbol{H}_{F,j}^t := \nabla^2 F_j(\boldsymbol{w}^t)$, and $\hat{\boldsymbol{H}}_{F,j}^t := \hat{\boldsymbol{H}}_{f,j}^t + \lambda\boldsymbol{I}$, where $\hat{\boldsymbol{H}}_{f,j}^t$ is the approximate matrix of $\nabla^2 f_j(\boldsymbol{w}^t)$ derived via the Nyström. We then obtain

$$\|e^t\| = \left\| (\boldsymbol{H}_F^t)^{-1} \bar{\boldsymbol{g}}^t - \frac{|\mathcal{D}_j|}{|\mathcal{D}|} \sum_{j=1}^m ((\hat{\boldsymbol{H}}_{F,j}^t)^{-1} \bar{\boldsymbol{g}}_j^t) \right\|$$

$$\leq \left\| \underbrace{\frac{|\mathcal{D}_j|}{|\mathcal{D}|} \sum_{j=1}^m \left( (\boldsymbol{H}_F^t)^{-1} - (\hat{\boldsymbol{H}}_{F,j}^t)^{-1} \right) \bar{\boldsymbol{g}}^t}_{e_1^t} + \underbrace{\frac{|\mathcal{D}_j|}{|\mathcal{D}|} \sum_{j=1}^m \left( (\hat{\boldsymbol{H}}_{F,j}^t)^{-1} (\bar{\boldsymbol{g}}^t - \bar{\boldsymbol{g}}_j^t) \right)}_{e_2^t} \right\|. \tag{17}$$

We first consider term $e_1^t$, where the second-order gradient error arises from both data heterogeneity and the Nyström approximation. By introducing the Hessian similarity bound inspired in (A2) of SCAFFOLD Karimireddy et al. (2020) , we obtain $\|\boldsymbol{H}_F^t - \boldsymbol{H}_{F,j}^t\| \leq \Gamma$. For any invertible matrices $\boldsymbol{A}$ and $\boldsymbol{A}'$, we have $(\boldsymbol{A}')^{-1} - \boldsymbol{A}^{-1} = \boldsymbol{A}^{-1}(\boldsymbol{A} - \boldsymbol{A}')(\boldsymbol{A}')^{-1}$. Then with Eq. equation 14 and $\delta \in (0,1)$, we obtain

$$\left\| (\boldsymbol{H}_F^t)^{-1} - (\boldsymbol{H}_{F,j}^t)^{-1} \right\| \leq \left\| (\boldsymbol{H}_F^t)^{-1} \right\| \left\| \boldsymbol{H}_F^t - \boldsymbol{H}_{F,j}^t \right\| \left\| (\boldsymbol{H}_{F,j}^t)^{-1} \right\| \leq \frac{\Gamma}{(1-\epsilon)\upsilon^2}, \tag{18}$$

with probability at least $1 - \delta$. Similarly, by applying equation 16 , we obtain

$$\left\| (\boldsymbol{H}_{F,j}^t)^{-1} - (\hat{\boldsymbol{H}}_{F,j}^t)^{-1} \right\| \leq \left\| (\boldsymbol{H}_{F,j}^t)^{-1} \right\| \left\| (\boldsymbol{H}_{F,j}^t)^{-1} - (\hat{\boldsymbol{H}}_{F,j}^t)^{-1} \right\| \left\| (\hat{\boldsymbol{H}}_{F,j}^t)^{-1} \right\|$$

$$\leq \frac{1}{(1-\epsilon)\upsilon} \rho_{Ny} \frac{1}{\lambda} \leq \frac{\rho_{Ny}}{(1-\epsilon)\upsilon\lambda}, \tag{19}$$

with probability at least $1 - 2\delta$, where $\|\hat{\boldsymbol{H}}_{F,j}^t\| = \|\hat{\boldsymbol{H}}_{f,j}^t + \lambda \boldsymbol{I}\| \succeq \lambda \boldsymbol{I}$ and the bound of $\|(\hat{\boldsymbol{H}}_{F,j}^t)^{-1}\|$ is derived as follows:

$$\left\| \hat{\boldsymbol{H}}_{F,j}^t \right\| \leq \left\| \boldsymbol{H}_{F,j}^t - \hat{\boldsymbol{H}}_{F,j}^t \right\| + \left\| \boldsymbol{H}_{F,j}^t \right\|$$

$$\leq (1+\epsilon)L + \rho_{Ny}. \tag{20}$$

Using Eq. equation 18 and Eq. equation 19, the error in the second-order gradient component can be bounded as:

$$\|e_1^t\| = \left\| \frac{|\mathcal{D}_j|}{|\mathcal{D}|} \sum_{j=1}^m \left( (\boldsymbol{H}_F^t)^{-1} - (\boldsymbol{H}_{F,j}^t)^{-1} + (\boldsymbol{H}_{F,j}^t)^{-1} - (\hat{\boldsymbol{H}}_{F,j}^t)^{-1} \right) \bar{\boldsymbol{g}}^t \right\|$$

$$\leq \frac{|\mathcal{D}_j|}{|\mathcal{D}|} \sum_{j=1}^m (\| (\boldsymbol{H}_F^t)^{-1} - (\boldsymbol{H}_{F,j}^t)^{-1} \| + \left\| (\boldsymbol{H}_{F,j}^t)^{-1} - (\hat{\boldsymbol{H}}_{F,j}^t)^{-1} \right\|) \|\bar{\boldsymbol{g}}^t\| \tag{21}$$

$$\leq \left( \frac{\Gamma}{(1-\epsilon)\upsilon^2} + \frac{\rho_{Ny}}{(1-\epsilon)\upsilon\lambda} \right) \|\bar{\boldsymbol{g}}^t\|,$$

with probability at least $1 - 2\delta$.

Next, we consider term $e_2^t$, which involves bounding the heterogeneity of the first-order gradients. To this end, we adopt the $B$-local dissimilarity from FedProxLi et al. (2020) (Definition 3). Under this bound, we obtain $\mathbb{E}[\|\bar{\boldsymbol{g}}_j^t\|^2] \leq B^2 \|\bar{\boldsymbol{g}}^t\|$, where $B > 1$. (It is worth noting that the bounds on first- and second-order local similarity are not directly related, so referencing different works does not affect the validity of our analysis. Moreover, the definitions of gradient similarity in these works are essentially equivalent, differing only in form—one expressed as an expected deviation, the other as an empirical deviation.) We define $\bar{\boldsymbol{g}}^{t,\text{avg}} := \frac{|\mathcal{D}_j|}{|\mathcal{D}|} \sum_{j=1}^m \bar{\boldsymbol{g}}_j^t$, so $\mathbb{E}[\bar{\boldsymbol{g}}^{t,\text{avg}}] = \bar{\boldsymbol{g}}^t$ and from them it follows that $\mathbb{E}[\bar{\boldsymbol{g}}^t - \bar{\boldsymbol{g}}^{t,\text{avg}}] = 0$. Based on the above, and by treating $\bar{\boldsymbol{g}}_j^t$ as independent across clients, we can derive:

$$\mathbb{E}[\|\bar{\boldsymbol{g}}^t - \bar{\boldsymbol{g}}^{t,\text{avg}}\|^2] = \mathbb{E}[\|\bar{\boldsymbol{g}}^{t,\text{avg}} - \mathbb{E}[\bar{\boldsymbol{g}}^{t,\text{avg}}]\|^2] = Var(\bar{\boldsymbol{g}}^{t,\text{avg}})$$

$$= Var(\frac{|\mathcal{D}_j|}{|\mathcal{D}|} \sum_{j=1}^m \bar{\boldsymbol{g}}_j^t) = \left( \frac{|\mathcal{D}_j|}{|\mathcal{D}|} \right)^2 \sum_{j=1}^m Var(\bar{\boldsymbol{g}}_j^t) = \frac{|\mathcal{D}_j|}{|\mathcal{D}|} Var(\bar{\boldsymbol{g}}_j^t) \tag{22}$$

$$\leq \frac{|\mathcal{D}_j|}{|\mathcal{D}|} \mathbb{E}[\|\bar{\boldsymbol{g}}_j^t\|^2] \leq \frac{|\mathcal{D}_j|}{|\mathcal{D}|} B^2 \|\bar{\boldsymbol{g}}^t\|^2,$$

where the last inequality uses the B-local dissimilarity. We further apply Markov's inequality to bound $\|\bar{\boldsymbol{g}}^t - \bar{\boldsymbol{g}}_j^t\|$ from above. Let $\epsilon_B > 0$ and $\delta \in (0, 1)$, when $B \leq \epsilon_B \delta \sqrt{\frac{|\mathcal{D}|}{|\mathcal{D}_j|}}$, we have

$$\|\bar{\boldsymbol{g}}^t - \bar{\boldsymbol{g}}_j^t\| \leq \epsilon_B \|\bar{\boldsymbol{g}}^t\|, \tag{23}$$

with probability at least $1 - \delta$. Consequently, we derive:

$$
\begin{aligned}
\|e_2^t\| &= \left\| \frac{|\mathcal{D}_j|}{|\mathcal{D}|} \sum_{j=1}^m \left( (\hat{\boldsymbol{H}}_{F,j}^t)^{-1} (\bar{\boldsymbol{g}}^t - \bar{\boldsymbol{g}}_j^t) \right) \right\| \\
&\leq \frac{1}{\lambda} \|\bar{\boldsymbol{g}}^t - \bar{\boldsymbol{g}}_j^t\| \\
&\leq \frac{\epsilon_B}{\lambda} \|\bar{\boldsymbol{g}}^t\|,
\end{aligned}
\tag{24}
$$

with probability at least $1 - 3\delta$, where the bound on $\|\hat{\boldsymbol{H}}_{F,j}^t\|$ has already been established in Equation Eq. equation 20.

Finally, we derive the following recurrence relation for $\|\boldsymbol{w}^t - \boldsymbol{w}^*\|$:

$$
\begin{aligned}
\|\boldsymbol{w}^{t+1} - \boldsymbol{w}^*\| &= \left\| \boldsymbol{w}^t - \boldsymbol{w}^* - \eta \frac{|\mathcal{D}_j|}{|\mathcal{D}|} \sum_{j=1}^m \left( (\hat{\boldsymbol{H}}_{F,j}^t)^{-1} \bar{\boldsymbol{g}}_j^t \right) \right\| \\
&= \left\| \boldsymbol{w}^t - \boldsymbol{w}^* - \eta (\boldsymbol{H}_F^t)^{-1} \bar{\boldsymbol{g}}^t + \eta e^t \right\| \\
&\leq \left\| \boldsymbol{w}^t - \boldsymbol{w}^* - \eta (\boldsymbol{H}_F^t)^{-1} \bar{\boldsymbol{g}}^t \right\| + \left\| \eta e^t \right\| \\
&\leq \|(\boldsymbol{H}_F^t)^{-1}\| \|\boldsymbol{H}_F^t(\boldsymbol{w}^t - \boldsymbol{w}^*) - \eta \bar{\boldsymbol{g}}^t\| + \left\| \eta e^t \right\| \\
&\leq \frac{1}{\upsilon} \left\| (\boldsymbol{H}_F^t - \boldsymbol{H}_F^*)(\boldsymbol{w}^t - \boldsymbol{w}^*) \right\| + \left\| \boldsymbol{H}_F^*(\boldsymbol{w}^t - \boldsymbol{w}^*) - \eta \bar{\boldsymbol{g}}^t \right\| + \left\| \eta e^t \right\| \\
&\overset{(i)}{\leq} \frac{1}{\upsilon} \left( M \left\| \boldsymbol{w}^t - \boldsymbol{w}^* \right\|^2 + (1 - \eta) \left\| \bar{\boldsymbol{g}}^t \right\| + \frac{M}{2} \left\| \boldsymbol{w}^t - \boldsymbol{w}^* \right\|^2 \right) + \left\| \eta e^t \right\| \\
&\leq \left( \frac{(1-\eta)L}{\upsilon} + \frac{\eta \Gamma L}{(1-\epsilon)\upsilon^2} + \frac{\eta \rho_{Ny} L}{(1-\epsilon)\upsilon \lambda} + \frac{\eta \epsilon_B L}{\lambda} \right) \left\| \boldsymbol{w}^t - \boldsymbol{w}^* \right\| \\
&\quad + \frac{3M}{2\upsilon} \left\| \boldsymbol{w}^t - \boldsymbol{w}^* \right\|^2,
\end{aligned}
\tag{25}
$$

with probability at least $1 - 3\delta$, where (i) is derived from Taylor expansion with integral remainder.

Choose the hyperparameters $\lambda > 4\eta \epsilon_B L$, $\frac{4L - \upsilon}{4L} < \eta < \min\{ \frac{(1-\epsilon)\upsilon^2}{4L\Gamma}, \frac{(1-\epsilon)\upsilon \lambda}{4(1+\epsilon_l)L^2} \}$, we can obtain:

$$
\begin{aligned}
P &= \frac{(1-\eta)L}{\upsilon} + \frac{\eta \Gamma L}{(1-\epsilon)\upsilon^2} + \frac{\eta \rho_{Ny} L}{(1-\epsilon)\upsilon \lambda} + \frac{\eta \epsilon_B L}{\lambda} \\
&\leq \frac{L}{\upsilon} \left( 1 - \frac{4L - \upsilon}{4L} \right) + \frac{(1-\epsilon)\upsilon^2 \Gamma L}{4L\Gamma(1-\epsilon)\upsilon^2} + \frac{(1-\epsilon)\upsilon \lambda \rho_{Ny} L}{4(1-\epsilon)\upsilon \lambda (1+\epsilon_l)L^2} + \frac{\eta \epsilon_B L}{4\eta \epsilon_B L} \leq 1.
\end{aligned}
\tag{26}
$$

The simplification of the third term follows from our definition in equation 16, $\rho_{Ny} = (1 + \epsilon_l)\lambda_{k+1}(\boldsymbol{H})$, $\lambda_{k+1}(\boldsymbol{H}) \leq \lambda_1(\boldsymbol{H}) \leq L$. Since the two parameter bounds contain mutually dependent terms, we substitute the upper bound $\frac{(1-\epsilon)\upsilon \lambda}{4(1+\epsilon_l)L^2}$ of $\eta$ into the lower bound of $\lambda$ to verify that the feasible set is non-empty. This yields $\frac{(1-\epsilon)\upsilon \epsilon_B}{(1+\epsilon_l)L} < 1$, which is readily satisfied under moderate heterogeneity, noting that $(1 + \epsilon_l) > (1 - \epsilon)$ and $L > \upsilon$. In addition, it can be observed that as heterogeneity increases (i.e., as $\epsilon_B$ and $\Gamma$ become larger), the admissible range for the hyperparameters becomes more restrictive, which is consistent with empirical observations.

$\square$

## C  PROOF OF THEOREM 2

*Proof.* This analysis assumes the initialization lies within a sufficiently small neighborhood of $\boldsymbol{w}^*$, which is standard in local convergence theory. Recall from equation 13 that we can rewrite the

expression in a simplified form:

$$\|\boldsymbol{w}^{t+1} - \boldsymbol{w}^*\| \le P\|\boldsymbol{w}^t - \boldsymbol{w}^*\| + Q\|\boldsymbol{w}^t - \boldsymbol{w}^*\|^2, \quad Q := \frac{3M}{2\upsilon} > 0. \tag{27}$$

Choose $\gamma > 0$ such that

$$\gamma \le \frac{1-P}{2Q}, \tag{28}$$

where $P < 1$ by Theorem 1. Define the convergence rate constant $\rho_r = P + Q\gamma$, which is bounded as:

$$\rho_r \le P + Q\frac{1-P}{2Q} = \frac{P+1}{2} < 1. \tag{29}$$

The convergence result holds under the assumption that the initial iterate satisfies $\|\boldsymbol{w}^0 - w^*\| \le \gamma$, where $\gamma \le \frac{1-P}{2Q}$ defines a local convergence region. We prove by induction that

$$\|\boldsymbol{w}^t - w^*\| \le \gamma\rho_r^t, \forall t \ge 0. \tag{30}$$

When $t = 0$, the inequality holds trivially as $\|\boldsymbol{w}^0 - \boldsymbol{w}^*\| \le \gamma = \gamma\rho_r^0$. Assume now that $\|\boldsymbol{w}^t - \boldsymbol{w}^*\| \le \gamma\rho_r^t$ holds for some $t > 0$. Then we have

$$\|\boldsymbol{w}^{t+1} - \boldsymbol{w}^*\| \le P\|\boldsymbol{w}^t - \boldsymbol{w}^*\| + Q\|\boldsymbol{w}^t - \boldsymbol{w}^*\|^2 \le P\gamma\rho_r^t + Q(\gamma\rho_r^t)^2. \tag{31}$$

Since $\rho_r^t \le 1$, it follows that $\gamma\rho_r^t \le \gamma$, and hence

$$Q(\gamma\rho_r^t)^2 = Q\gamma^2\rho_r^{2t} \le Q\gamma^2\rho_r^t. \tag{32}$$

Substituting yields

$$\|\boldsymbol{w}^{t+1} - \boldsymbol{w}^*\| \le P\gamma\rho_r^t + Q\gamma^2\rho_r^t = \gamma\rho_r^t(P + Q\gamma) = \gamma\rho_r^{t+1}, \tag{33}$$

where the last equality uses the definition $\rho_r = P + Q\gamma$. This proves equation 30.

To achieve $\|\boldsymbol{w}^t - w^*\| \le \varepsilon$, it suffices to ensure

$$\begin{aligned} \gamma\rho_r^T &\le \varepsilon, \\ T\log\frac{1}{\rho_r} &\ge \log\frac{\gamma}{\varepsilon}, \\ T &\ge \frac{\log(\gamma/\varepsilon)}{\log(1/\rho_r)}, \end{aligned} \tag{34}$$

where the second inequality follows from the fact that $\rho_r < 1$. Since $\gamma$ and $\rho_r$ are constants independent of $\varepsilon$, the iteration complexity is

$$T = \mathcal{O}(\log\frac{1}{\varepsilon}). \tag{35}$$

$\square$

## D EXPERIMENTAL SETTING

We provide detailed experimental settings and parameter choices used in our evaluations. For general hyperparameters including learning rate $\eta$ and regularization parameter $\lambda$, we report representative ranges in the table 4 for clarity across the four models, except for FedNL and DONE, whose settings are provided later. We then present method-specific configurations and experimental details with fixed parameter values. All experiments for our proposed method were conducted on a single NVIDIA RTX 4090 GPU.

- **FN-NOW.** We set the singular value regularization parameter $\lambda_s = 10^{-4}$ in all experiments. For MLR, MLP, and CNN, we use sampling rates of 0.003, 0.0003, and 0.00008, respectively, while for ResNet, we directly set $d = 1$.
- **Fed-Sophia.** For ADAM-like momentum parameters, we set $(\beta_1, \beta_2)$ to (0.95, 0.99) for MLR, (0.90, 0.99) for MLP, and (0.90, 0.95) for CNN and ResNet. Under partial client participation setting, we reduce them to (0.50, 0.55). The Hessian clipping threshold is set to $10^{-4}$.

Table 4: The parameter $\eta$, $\lambda$ value ranges of each method under different experimental settings in this study.

| DISTRIBUTION | | FEDAVG | SCAFFOLD | FED-SOPHIA | FAGH | **FN-NOW** |
|---|---|---|---|---|---|---|
| IID | $\eta$ | [1E-2, 7E-1] | [5E-3,E3-1] | [1E-4,8E-3] | [1E-4,8E-3] | [8E-4,5E-3] |
| | $\lambda$ | [1E-2,1.5E-2] | [1E-3,1E-2] | [4.5E-3,1E-2] | [2E-3,8E-2] | [1.5E-3,1E-1] |
| $\alpha = 0.5$ | $\eta$ | [8E-3, 4E-1] | [7E-3,1.2E-1] | [1.5E-4,9E-4] | [1E-4,5E-2] | [8E-4,3E-3] |
| | $\lambda$ | [1E-2,1.2E-2] | [5E-3,3E-2] | [4.5E-3,5E-3] | [1E-2,8E-2] | [5E-3,1E-2] |
| $\alpha = 0.1$ | $\eta$ | [5E-3, 5E-2] | [5E-3,9E-2] | [7E-5,8E-4] | [5E-4,6E-3] | [8E-4,5E-2] |
| | $\lambda$ | [1E-2,1.5E-2] | [1E-3,1E-2] | [4.5E-3,5E-3] | [8E-3,3E-2] | [1E-3,1E-2] |

- **FedNL.** Chaudhuri et al. discusses several variants, among which we adopt the vanilla version of FedNL. After comparing Unbiased Compressors, Contractive Compressors, and Low-rank Compressors, we find that using option 2 combined with a rank-20 low-rank compression yields the best performance. Other hyperparameters are set to $\eta = 1$ and $\alpha = 1$.

- **DONE.** We set $R = 40$ and $\alpha = 0.03$ (as used in the DONE algorithm). When the Dirichlet parameter is 0.5, we use $\eta = 0.02$ and $\lambda = 0.005$; for all other cases, we set $\eta = 0.03$ and $\lambda = 0.001$.

# E EXPERIMENTAL SUPPLEMENT

## E.1 COMPARISON ON CNN

The result in Figure 8 supplements the baseline comparisons in overall training performance and the experimental setup and analysis follow the main text.

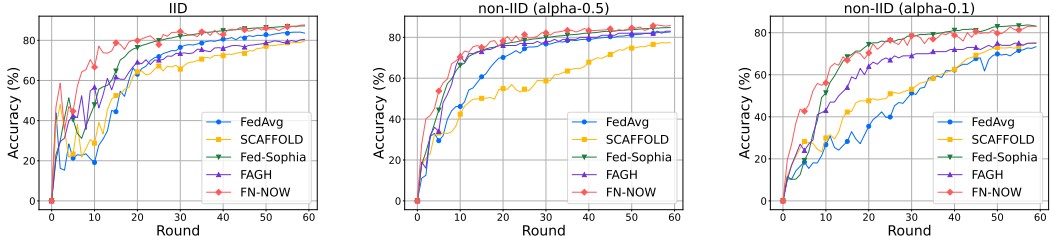

Figure 8: The test accuracy of the compared methods on Fashion MNIST using 5-layer CNN under different levels of data heterogeneity.

## E.2 PARTIAL PARTICIPATION SETTING

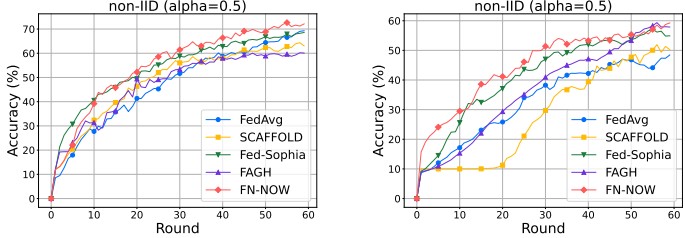

Figure 9: Test accuracy on non-IID on non-IID ($\alpha = 0.5$) MNIST (left) and Fashion MNIST (right) using MLP and a 5-layer CNN, respectively, under partial client participation.

Under moderate heterogeneity, we evaluate partial client participation across models and datasets. We use 100 clients and sample 30 at random per round. As shown in Figure 9, all methods require more rounds to converge than under full participation, yet our method remains among the fastest. On CNNs, SCAFFOLD exhibits a pronounced slow start, likely because its control variate correction cannot reflect a reliable global direction when only a subset of clients has participated early on. Similarly, methods such as Fed-Sophia that reference previous client updates benefit from reducing reliance on prior rounds, typically requiring a lower momentum parameter.

### E.3 Additional Results on Communication Efficiency

In Table 5, we provide the comparative data on communication cost and the number of training rounds required to achieve a specific accuracy using CNN and MLR, which was omitted in the main text. The communication cost refers to the amount of data transmitted in each round, which remains fixed for a given method and model throughout training. Notably, FedNL and DONE rely on computing the full Hessian matrix, which restricts their applicability to models with a relatively small number of parameters like CNN due to memory constraints. FAGH and SCAFFOLD exhibit nearly identical communication efficiency since both methods transmit two parameter-sized vectors per round; the slight differences observed are likely due to statistical variation. DONE incurs this communication overhead because it performs two communication steps per round. Although FedNL employs compression techniques to avoid transmitting the full quadratic-size Hessian matrix (e.g., 7035.08MB per round in the MLR experiment), it still incurs significantly higher per-round cost compared to other methods. In contrast, our method achieves the same per-round communication cost as FedAvg while requiring substantially fewer rounds to converge.

Table 5: The comparison of the number of communication cost (Comm.) and rounds required by the compared methods to achieve a target accuracy(%) using CNN and MLR.

| Method | Target Accuracy - CNN | | | | | | Target Accuracy - MLR | | | | | |
| | Comm. (MB) | IID | | $\alpha = 0.5$ | | $\alpha = 0.1$ | | Comm. (MB) | IID | | $\alpha = 0.5$ | | $\alpha = 0.1$ | |
| | | 70 | 80 | 70 | 80 | 65 | 75 | | 80 | 85 | 80 | 85 | 80 | 85 |
|---|---|---|---|---|---|---|---|---|---|---|---|---|---|---|
| FedAvg | 134.55 | 23 | 38 | 19 | 44 | 43 | - | 0.90 | 9 | 29 | 17 | 38 | 18 | 41 |
| SCAFFOLD | 278.62 | 34 | - | 43 | - | 41 | 57 | 1.80 | 34 | - | 22 | 74 | 28 | - |
| FedNL | \ | \ | \ | \ | \ | \ | \ | 36.79 | - | - | 43 | - | - | - |
| DONE | \ | \ | \ | \ | \ | \ | \ | 1.79 | 29 | 49 | 30 | 58 | 40 | 59 |
| Fed-Sophia | 134.55 | 15 | 25 | 11 | 30 | 13 | **20** | 0.90 | 6 | 31 | **5** | **10** | 16 | 72 |
| FAGH | 278.38 | 21 | 57 | **9** | 37 | 20 | 49 | 1.79 | 9 | 17 | 22 | - | - | - |
| **FN-NOW** | 134.55 | **8** | **18** | **9** | **20** | **11** | 23 | 0.90 | **3** | **26** | 8 | 14 | **9** | **25** |

### E.4 Additional Results on Model Accuracy

Table 6: The summary of final-round accuracy (%) for the compared methods using MLR.

| Method | IID | $\alpha = 0.5$ | $\alpha = 0.1$ |
|---|---|---|---|
| FedAvg | 88.08±0.00 | 87.96±0.45 | 88.06±0.24 |
| SCAFFOLD | 84.56±0.05 | 85.81±0.23 | 83.99±0.09 |
| FedNL | 67.05±2.03 | 57.53±6.73 | 56.76±0.85 |
| DONE | 88.41±0.14 | 88.25±0.16 | **88.59±0.14** |
| Fed-Sophia | 91.52±0.23 | 89.39±0.68 | 85.39±0.40 |
| FAGH | 89.24±0.09 | 78.44±0.11 | 68.52±0.08 |
| **FN-NOW** | **91.98±0.02** | **89.89±0.14** | 88.46±0.12 |

We report the final-round accuracies on MLR in Table 6, with the round fixed at 60. As shown, our method achieves the highest accuracy under both the IID and $\alpha = 0.5$ settings, and remains competitive under $\alpha = 0.1$. As previously discussed, MLR has relatively few parameters, making it well-suited to first-order methods. In this regime, introducing complex approximations or second-order information may hinder training performance rather than improve it. For example, FedNL

applies compression techniques, and FAGH constructs second-order updates based on the first row of the Hessian, both of which may introduce noise or information loss. Moreover, we observe that DONE appears relatively insensitive to data heterogeneity, likely because it performs local updates guided by a shared global gradient. Overall, although our method is primarily designed to improve communication efficiency, it maintains strong performance without sacrificing training effectiveness.

# F  THE USE OF LARGE LANGUAGE MODELS (LLMS)

We used large language models solely for writing assistance in polishing phrasing and correcting spelling and grammar. The authors remain fully responsible for the paper's accuracy and integrity.

