# OpenReview forum: "FN-NOW: A Communication-Efficient Newton-Type Federated Learning via Low-Rank Hessian Approximation"
_ICLR.cc/2026/Conference — ICLR 2026 Conference Withdrawn Submission_

### Official Review · Reviewer_8o6z · 2025-10-19

**Soundness:** 2
**Presentation:** 2
**Contribution:** 2
**Rating:** 4
**Confidence:** 3

**Summary:**

This paper proposes FN-NOW, a federated learning (FL) algorithm that combines Newton-type optimization with low-rank Hessian approximation to achieve communication efficiency. The key innovation is using the Nyström method to approximate the Hessian matrix and the Woodbury identity to avoid explicit reconstruction of large matrices, enabling communication costs comparable to first-order methods while retaining theoretical convergence guarantees. The authors provide convergence analysis showing linear convergence rates and demonstrate experimental improvements over existing baselines on MNIST, Fashion-MNIST, and CIFAR-10 across multiple model architectures.

**Strengths:**

- The paper addresses a genuine challenge in federated learning, i.e., reconciling the computational advantages of second-order methods with their prohibitive communication and memory costs. This is an important practical problem.
- Providing a convergence proof (Theorem 1-2) with a linear convergence rate is valuable.
- The paper addresses numerical stability through singular value regularization with an explicit stabilization parameter $\lambda_s$, showing awareness of implementation challenges.
- Figure 1 provides a clear visual breakdown of the algorithm steps, and Algorithm 1 is reasonably well-specified.

**Weaknesses:**

- Theorem 1 requires twice-differentiable convex functions (Assumption 1), but all experiments use neural networks which are non-convex.
- The convergence guarantee requires initialization within a ball (Theorem 2), which is a strong requirement. No discussion of how restrictive this is in practice or how to achieve it.
- The paper claims linear convergence but the rate constant $P$ depends on multiple unknown quantities ($\Gamma$, $\epsilon_B$) whose magnitudes are never empirically characterized. It's unclear whether linear convergence manifests in realistic settings.
- Nyström approximation of Hessians is well-established. The Woodbury identity application is straightforward, given the low-rank structure. The main contribution is combining these existing techniques rather than introducing fundamentally new ideas.
- Leverage score sampling (Eq. 8) is based on squared gradients, which is a simple heuristic without strong justification. Why not use the actual leverage scores of the Hessian?
- The per-round communication is claimed to be $\mathcal{O}(D)$, matching first-order methods. However, the method requires sampling and computing Hessian subsets locally, which involves gradient computations for d dimensions. The per-device computation cost could still be higher than claimed.
- Figure 6 shows wall-clock time comparisons but uses a fixed 10 Mbps link rate assumption. Real network conditions vary significantly.
- The connection between local Hessian heterogeneity and the error term $\epsilon_B$ is not intuitive. The paper references FedProx's $B$-dissimilarity but doesn't explain why this same bound applies to Hessians.
- No ablation study on the Nyström + Woodbury combination. What if only one is used?
- Sampling strategy (leverage scores vs. uniform vs. random) is not compared empirically.
- Recent work on quasi-Newton methods and natural gradient in FL could be discussed.
- Some implementation details are missing: How is the SVD computed for $\tilde{H}_{dd}$ in practice? Is a rank truncation applied?
- In the FedNL reference, the author names are incorrect.

**Questions:**

Refer to the weaknesses above.

---

### Official Review · Reviewer_YMnL · 2025-10-24

**Soundness:** 3
**Presentation:** 3
**Contribution:** 1
**Rating:** 4
**Confidence:** 4

**Summary:**

The paper proposes FN-NOW which is a Newton-type local optimization method in federated learning. The algorithm replaces the full Hessian with a low-rank Nyström approximation built from a subset of gradients (using a leverage-score sampling rule and using gradient norm as the score), and computes the inverse of the approximate Hessian efficiently through the Woodbury identity, keeping computation overhead low.

The paper provides a convergence analysis under convex assumptions and experiments on MNIST, Fashion-MNIST, and CIFAR-10, showing faster convergence and lower communication cost compared with first-order and other second-order baselines.

**Strengths:**

This is a narrow work that applies second-order optimization in local training in federated learning. The algorithm is very clear and well written. Using Woodbury identity is a good idea to efficiently compute inverses.
The convergence analysis is concrete.
FN-NOW shows faster convergence and better accuracy than baselines like FedAvg, SCAFFOLD, Fed-Sophia, and FAGH on standard datasets and models.

**Weaknesses:**

Experiment settings are unclear. For example, local epochs (which appear to be implicitly set to 1 based on Algorithm 1, but this is not stated outright), batch sizes are missing.

The evaluation methodology raises concerns by enforcing a fixed number of communication rounds across all baselines (as stated around line 462). It is unfair to compare the accuracies of first-order and second-order methods given that each algorithm might require a different number of rounds.

Experiments are limited. Considering that a second-order optimizer adapts to each local optimization more quickly, and that stronger local adaptation may lead to larger client drift (or discrepancy), it is unclear how FN-NOW behaves for larger and more challenging datasets.
The applicability to massive models like large language models (LLMs) is untested and uncertain. FN-NOW's Hessian-based approach may face prohibitive computational or memory costs for billion-parameter models, especially in resource-constrained FL environments.
Theoretical results depend on restrictive assumptions like strong convexity, Lipschitz Hessians, and assumptions to make P<1.

**Questions:**

The assumptions (including line 790) for making “P<1” in the convergence analysis seems quite strong. Could you elaborate how likely are they to hold in practice?
Do the authors have any arguments about why FN-NOW beats Fed-Sophia? I think leverage-score sampling is an important step that makes FN-NOW works better since it might work as implicit regularization.
Do the authors have any justification about using the same fixed number of training rounds for all methods?

---

### Official Review · Reviewer_FqKp · 2025-10-27

**Soundness:** 2
**Presentation:** 2
**Contribution:** 1
**Rating:** 2
**Confidence:** 5

**Summary:**

This paper introduces FN-NOW, a communication-efficient second-order algorithm based on low-rank approximation of the Hessian. In particular, using the Nyström method and the Woodbury identity, FN-NOW computes the Hessian's inverse more efficiently. The authors provide a convergence analysis of the proposed method along with the experimental tests that demonstrate the competitive performance in practice.

**Strengths:**

- The authors test the performance of the proposed algorithm in training MLP on MNIST and Resnet18 on CIFAR10.

**Weaknesses:**

- Wrong references: last line of page 6 (reference to DONE algorithm, not FedProx); FedNL algorithm has wrong author list. There are other problems with the references (e.g., "just accepted" for FedRL paper looks inappropriate to me). Therefore, I encourage the authors to clean up all the references.

- There is a restriction on the local dataset size $|D_j|$ and the sampling dimension of the problem $d$ in the convergence analysis, i.e., they should be large enough.

- It is unclear whether the constant $P$ in the convergence analysis can be made small, especially for the problem with large conditioning $L/v$. In particular, $P=(1-\eta)L/v + \eta\Gamma L/((1-\epsilon)v)+...$. If $L/v$ is large, then we need to choose $\eta$ to be close to $1$ to make the first term smaller than $1$. However, in this case, the second term becomes large. Therefore, I am not sure that the condition $P< 1$ can be satisfied, which is necessary for the convergence analysis.

- The authors did not provide a full list of assumptions in the main paper, which significantly limits the readability. For example, the authors refer to constants $\Gamma$ and $\epsilon$ as some sort of local similarity, but fail to provide their definition. Also, the authors use strong growth condition in line 743, which is also not mentioned in the main paper.

- It looks suspicious to me that first-order algorithms like SCAFFOLD that are known to achieve a high performance in practice fail to converge according to the results in Table 2. This questions the experimental part of this work; the baselines might have suboptimal performance due to insufficient tuning (or other reasons). I am also concerned that the hyperparameters that are mentioned in lines 876-883 are well justified. For example, in the FedNL paper, the authors suggest using Rank-1 compression and option 1, while the authors here use Rank-20 and option 2.

- A proper high-probability analysis is required to demonstrate the convergence analysis of FN-NOW. In the current form, I don't see why the probabilities are summed, not multiplied.

- I am not sure that the condition on the stepsize $\eta$ in line 790 can be satisfied, again because the condition number $L/v$ is typically large. Therefore, the LHS $\frac{4L-v}{4L}$ is close to 1, while the RHS $\frac{(1-\epsilon)v^2}{L\Gamma}$, in opposite, close to $0$ for the same reason.


Considering the amount of unclear statements, most likely mistakes/wrong claims in the proofs, unsatisfactory writing (not listing all assumptions), and other issues of this work, I recommend rejecting without any possibility for increasing the score.

**Questions:**

- What is the definition of $H_dd$?

- Why is $\frac{|D_j|}{|D|}$ outside of the sum in (6)? There are many places where this typo/mistake appears. I believe the authors implicitly assume that the ratio does not depend on $j$.

- From (5), $F$ is already $\lambda$-strongly convex. Why do the authors use assumption 1? Is there any necessity to separate the strong convexity parameter $v$ from $\lambda$?

- What is small $c$ in Theorem 1 and Lemma 1? How is it defined? Why do we need it? Can we choose $c=1$? How does the choice of this constant affect the algorithm's convergence analysis and performance?

- What are $\Gamma$ and $\epsilon_B$ in Theorem 1? Where can I find their definitions? What is "local similarity" used in Theorem 1?

- What are first and second order gradients?

- The authors use "linear convergence, as typical for Newton-type methods". In my view, this sentence is misleading as second-order algorithms are expected to have a superlinear rate, not just linear.

- What is meant by "loss of Hessian information" in line 418?

- $H_F^*$ in line 780 is not defined.

---

### Official Review · Reviewer_pDYn · 2025-10-29

**Soundness:** 1
**Presentation:** 2
**Contribution:** 2
**Rating:** 2
**Confidence:** 4

**Summary:**

This paper introduces FN-NOW, a communication-efficient Newton-type federated learning algorithm that employs low-rank Hessian approximation using the Nyström method and Woodbury identity. The goal is to retain second-order convergence speed while reducing the computational, memory, and communication overhead typical in Newton-type FL. Theoretical analysis establishes linear convergence under convex smoothness assumptions, and experiments demonstrate higher accuracy than first- and second-order FL baselines.

**Strengths:**

1. The paper identifies an interesting field — the high overhead of second-order methods in FL — and provides a computationally efficient solution via the Nyström–Woodbury combination.
2. The convergence proof shows linear convergence and reduces communication and memory cost, making second-order FL more practical.
3. Multiple datasets, architectures, and heterogeneity settings are evaluated, showing consistent improvement.

**Weaknesses:**

1. Sampling only a small subset of Hessian columns inevitably introduces curvature bias, causing the method’s claimed “second-order behavior” to deviate from true Newton directions. The paper neither discusses this bias nor provides any mechanism to mitigate it.
2. The so-called “leverage scores” are defined by squared gradient magnitudes rather than true Hessian-based leverage scores. The paper offers no theoretical justification that these probabilities lead to faithful Hessian approximation. Moreover, this approach can be unstable when feature scales differ, as large-magnitude gradients may dominate sampling, introducing bias and noise.
3. While the Woodbury identity effectively reduces computational complexity by approximating the inverse Hessian, this technique is not novel. It is commonly used in bilevel optimization to approximate inner-level Hessian inverses. FN-NOW neither discusses this connection nor provides new theoretical insights beyond standard usage. Furthermore, the paper does not analyze the approximation bias introduced by replacing exact Hessian inverses, nor does it study how the Woodbury bias interacts with federated heterogeneity, where each client applies a separate low-rank decomposition.
4. The convergence analysis assumes convex and smooth objectives, while the experiments are conducted on non-convex neural networks, leaving a clear gap between theoretical guarantees and practical settings.
5. Although non-IID experiments are included, the theoretical analysis lacks a formal definition or modeling of client heterogeneity.
6. Some minor typoes, such as:
  - line 212 "we use leverage";
 -  line 427 "communication efficiency" should begin with capital letters.

**Questions:**

See weaknesses above.

---

### Note · Authors · 2025-11-12

I have read and agree with the venue's withdrawal policy on behalf of myself and my co-authors.